# Reconfigurable hyperbolic polaritonics with correlated oxide metasurfaces

Neda Alsadat Aghamiri [1], Guangwei Hu [2,3], Alireza Fali[1], Zhen Zhang[4], Jiahan Li [5], Sivacarendran Balendhran [6], Sumeet Walia [7,8], Sharath Sriram[8,9], James H. Edgar [5], Shriram Ramanathan[4], Andrea Alù [2,10] & Yohannes Abate [1] ✉

Polaritons enable subwavelength confinement and highly anisotropic flows of light over a wide spectral range, holding the promise for applications in modern nanophotonic and optoelectronic devices. However, to fully realize their practical application potential, facile methods enabling nanoscale active control of polaritons are needed. Here, we introduce a hybrid polaritonic-oxide heterostructure platform consisting of van der Waals crystals, such as hexagonal boron nitride (hBN) or alpha-phase molybdenum trioxide (α-$MoO_3$), transferred on nanoscale oxygen vacancy patterns on the surface of prototypical correlated perovskite oxide, samarium nickel oxide, $SmNiO_3$ (SNO). Using a combination of scanning probe microscopy and infrared nanoimaging techniques, we demonstrate nanoscale reconfigurability of complex hyperbolic phonon polaritons patterned at the nanoscale with high resolution. Hydrogenation and temperature modulation allow spatially localized conductivity modulation of SNO nanoscale patterns, enabling robust real-time modulation and nanoscale reconfiguration of hyperbolic polaritons. Our work paves the way towards nanoscale programmable metasurface engineering for reconfigurable nanophotonic applications.

Polaritons are hybrid light-matter particles that offer promise to manipulate light at subwavelength scales[1–3]. Advances in hybridizing polaritonic materials with optically active systems and correlated oxides have recently attracted widespread attention due to their ability to achieve functionalities that are actively tunable. For example, tunable polaritonic metasurfaces have been pursed for reconfigurable nano-optic functionalities in compact devices[4,5]. One strategy is to utilize phase-change media as a substrate to reconfigure van der Waals nanomaterials, thus modulating the supported phonon polaritons

(PhPs)[4]. For instance, hyperbolic PhPs (HPhPs), featuring an anisotropic dispersion that results in hyperbolic iso-frequency contours associated with associated large light–matter interactions can be manipulated with phase change substrates[4,6,7]. Tunable polariton generation that relies on temperature[8–11] modulation typically introduces loss, as the phonon lifetime is reduced when the temperature, and correspondingly the phonon–phonon scattering events, increase. In addition, this route to polariton tunability suffers from inaccessibility of nanoscale manipulation due to the inability to design localized

[1]Department of Physics and Astronomy, University of Georgia, Athens, GA 30602, USA. [2]Photonics Initiative, Advanced Science Research Center, City University of New York, New York, NY 10031, USA. [3]Department of Electrical and Computer Engineering, National University of Singapore, Kent Ridge, Singapore 117583, Singapore. [4]School of Materials Engineering, Purdue University, West Lafayette, IN 47907, USA. [5]Tim Taylor Department of Chemical Engineering, Kansas State University, Manhattan, KN 66506, USA. [6]School of Physics, University of Melbourne, Parkville, VIC 3010, Australia. [7]School of Engineering RMIT University Melbourne, Melbourne, VIC, Australia. [8]Functional Materials and Microsystems Research Group and the Micro Nano Research Facility RMIT University, Melbourne, VIC, Australia. [9]ARC Centre of Excellence for Transformative Meta-Optical Systems, RMIT University, Melbourne, VIC, Australia. [10]Physics Program, Graduate Center, City University of New York, New York, NY 10016, USA. ✉e-mail: yohannes.abate@uga.edu

thermal manipulation. As a result, reconfigurability of nanopolaritonic structures remains limited.

Correlated oxides provide opportunities to reconfigure polaritonic devices at the nanoscale, due to their highly tunable optical and electronic properties[12–14]. Among these, rare-earth nickelates ($RNiO_3$, R = rare-earth element) display a rich phase diagram of structural and physical properties[15–18], controllable through various approaches, including doping[19,20], temperature[21], atomic vacancies[22], electric bias[23], and more. Hence, they have been investigated for reconfigurable nanoelectronics, fuel cells, and memristor devices[20,23–26]. As an example, the functional properties of $SmNiO_3$ (SNO) are sensitive to the orbital occupancy of electrons via carrier doping[23,27,28], exhibiting a giant change—more than eight orders of magnitude—in resistivity, as well as an order of magnitude in optical bandgap at its hydrogen-doping driven Mott transition[28]. These properties enable SNO to reversibly change its refractive index over a broad frequency range, which has been explored for nanophotonic applications including electro-optic modulators for controllable scattering[28], while other opportunities remain unveiled.

Here, we demonstrate correlated oxide polaritonic metastructures with on-demand and multimode programming of the supported polaritons at the nanoscale, achieved through field generated oxygen vacancies, hydrogen doping, as well as temperature modulation. We first present the characterization and manipulation of optical properties of SNO, then we demonstrate case studies of two prototypical hyperbolic van der Waals crystals, hexagonal boron nitride (hBN)[29–31] and alpha-phase molybdenum trioxide ($\alpha$-$MoO_3$)[32–34], showcasing rich dispersion tuning of HPhPs and enabling diverse control and patterning of PhPs. Our results reveal unique opportunities for real-time nanoscale tunability of nanophotonic devices, thus advancing reconfigurable and programmable nanophotonic technologies.

## Results

### Tailoring the local conductivity of SNO ($SmNiO_3$)

We consider nanostructured surfaces based on SNO grown by magnetron co-sputtering on a $LaAlO_3$ substrate, and subsequently annealed in high pressure oxygen gas (see Methods, material synthesis for the detailed process). We first experimentally demonstrate multimodal hyperbolic metasurfaces by active lateral manipulation of electronic phases of SNO. To this end, we combine conductive atomic force microscopy (c-AFM) and scanning Kelvin probe microscopy (SKPM) (with the Cypher AFM from Oxford Instruments) to generate and characterize patterns of various levels of conductivities on SNO. To write square charge patterns on a SNO sample (topography shown in Fig. 1b), we applied strong local fields via c-AFM operated in contact mode, detection set point 0.2 V, scan rate 0.5 Hz, and spring constant of the tip 2 N$m^{-1}$. Subsequently, we mapped the surface potential of the written areas using SKPM operated in tapping mode with an amplitude of 500 mV (see "Methods", SKPM). The positive bias voltage in all

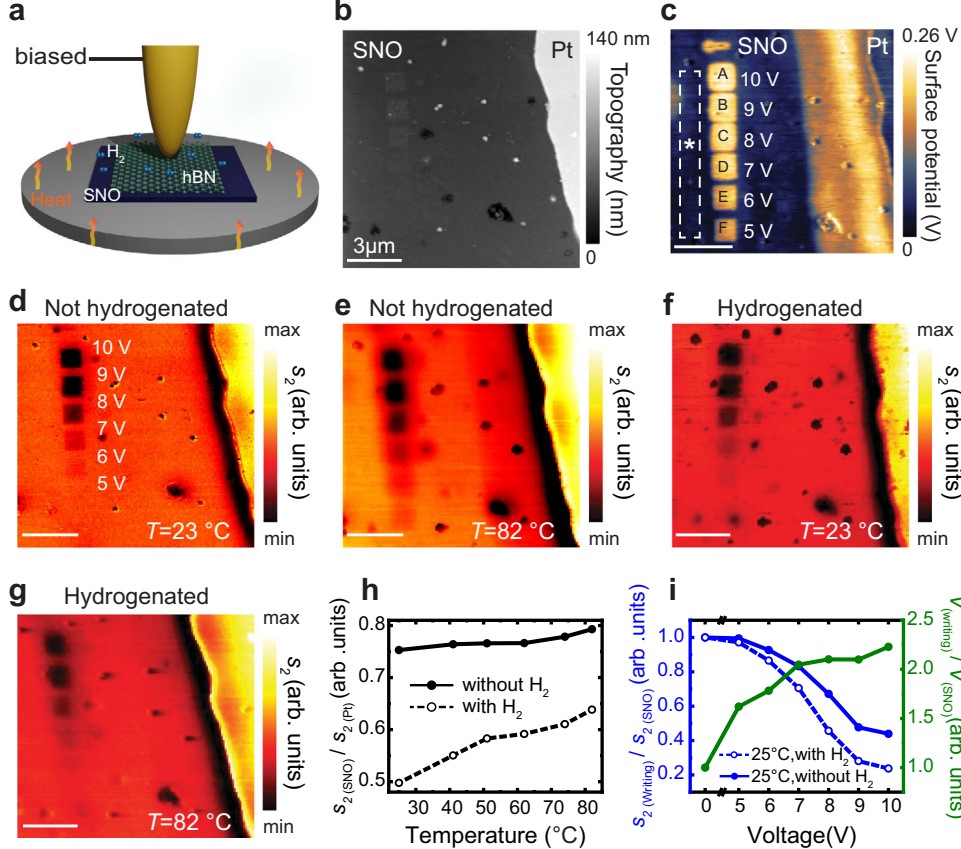

**Fig. 1 | Tailoring the local conductivity of SNO ($SmNiO_3$). a** Schematics of integrated s-SNOM (Scattering type Scanning Near-field Optical Microscopy) and heating setup, in case of conductive writing tip is biased. **b** AFM (Atomic Force Microscopy) topography image of pristine SNO. **c** SKPM (Scanning Kelvin Probe Microscopy) surface potential image of conductive writing patterns on pristine SNO made by applying 5–10 V potential at the c-AFM (Conductive AFM) tip, corresponding infrared s-SNOM second harmonic near-field amplitude $s_2$ images at **d** $T = 23\,°C$ (no $H_2$ exposure), **e** $T = 82\,°C$ (no $H_2$ exposure), **f** $T = 23\,°C$, after exposing to 5% $H_2$ gas at 100 °C, **g** $T = 82\,°C$, after exposing to 5% $H_2$ gas at 100 °C. All samples were imaged at $\lambda = 10.5\,\mu m$. **h** Normalized amplitude plots with and without $H_2$ exposure. **i** Normalized amplitude plots with and without $H_2$ exposure at $T = 25\,°C$ (blue lines) and normalized potential plot of regions A–F (green line) as a function of voltage. All data points of plots were taken at regions with different potential (A–F) shown in panel (**c**), and reference to normalize amplitude data is taken at * in panel (**c**). Scale bars indicate 3 $\mu m$.

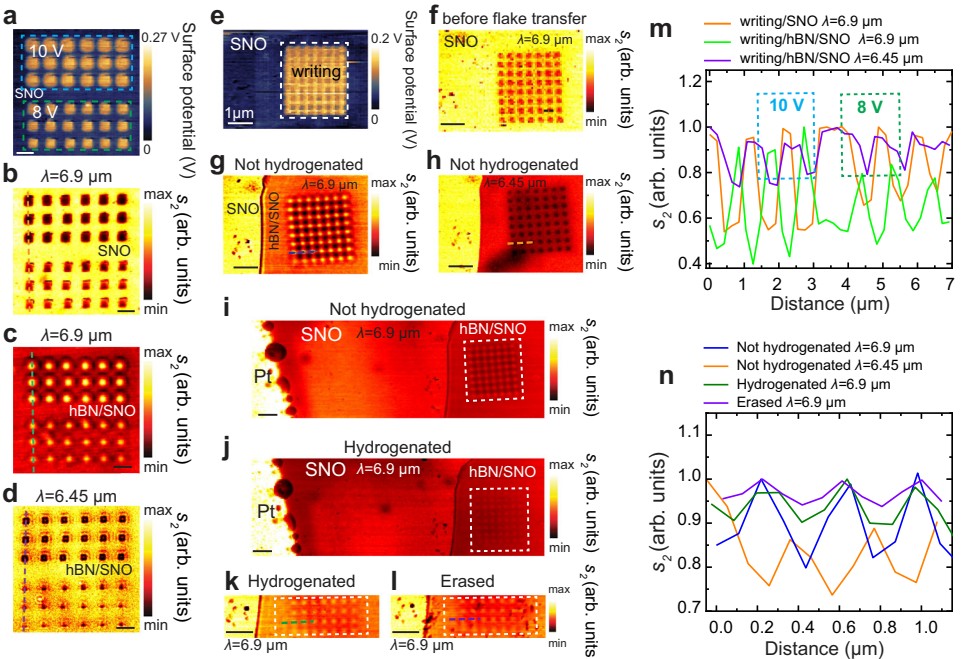

**Fig. 2 | Nanophotonic design of HPhPs structures on hBN via controllable and tunable localized conductivities of the SNO substrate. a** SKPM surface potential image of pristine SNO with conductive writing patterns made by applying potential at the c-AFM tip, 10 V (three top rows) and 8 V (three bottom rows). **b** Corresponding s-SNOM second harmonic near-field amplitude, $s_2$, image of panel (**a**) taken at $\lambda$ = 6.9 μm excitation wavelength. S-SNOM second harmonic near-field amplitude, $s_2$, image of panel (**a**) with a hBN flake transferred on top of the pattern imaged at **c** $\lambda$ =6.9 μm excitation wavelength and at **d** $\lambda$ = 6.45 μm. **e** SKPM surface potential image of pristine SNO with conductive writing patterns made by applying 10 V potential at the c-AFM tip. **f** Corresponding s-SNOM second harmonic near-field amplitude, $s_2$, image of panel e taken at $\lambda$ = 6.9 μm excitation wavelength. S-SNOM second harmonic near-field amplitude, $s_2$, image of panel (**a**) with a hBN

flake transferred on top of the pattern imaged at **g** $\lambda$ = 6.9 μm excitation wavelength and at **h** $\lambda$ = 6.45 μm. **i** Near-field second harmonic amplitude, $s_2$, images of the conductive pattern on hBN/SNO and Pt electrode shown on the left of the patterns before $H_2$ exposure **i**, after $H_2$ exposure **j**, zoomed in image after $H_2$ exposure **k** followed by erasing made by applying 10 V potential at the c-AFM tip (**l**). **m** Near-field second harmonic amplitude, $s_2$, line profiles extended along the dashed lines in panels **b–d** (orange, neon green, and violet, respectively). **n** Near-field second harmonic amplitude, $s_2$, line profiles extended along the dashed lines in panels **g**, **h**, **k**, and **l** (blue, orange, green, and violet respectively). White dashed lines in panels **e**, **i**, **j**, **k**, and **l** represent the charge written region on SNO. All scale bars indicate 1 μm.

---

writings leads to a large a potential compared to pristine SNO, as can be appreciated in the bright images in Fig. 1c. A biased tip enables oxygen vacancy generation on the SNO surface, providing a controllable route to manipulate the surface conductivity of SNO[35]. This charge writing behavior is reversible with a negative bias[35]. Figure 1c shows the potential distribution of the square writings ($V_{\text{(writing)}}$) normalized to the potential of pristine SNO ($V_{\text{(SNO)}}$), showing how the surface potential difference of the patterned areas increases with the tip bias voltage, commensurate with the change in local conductivity. In order to quantitatively assess the local conductivity modulation of the charge writing process, we performed scattering type canning near-field microscopy (s-SNOM) imaging of the patterned area of the sample. S-SNOM enables imaging local conductivity changes with high-sensitivity and high-spatial resolution, limited only by the sharpness of the probe tip (see "Methods")[36–39]. The resulting s-SNOM amplitude images shown in Fig. 1d−g were taken at 10 μm laser wavelength and show the voltage-dependent optical contrast. Such s-SNOM image contrast directly reflects local conductivity changes[38] of the sample where dark regions, i.e., the area written with positive biases, imply lower conductivity compared to the pristine (red) region (Fig. 1d−g) due to the removal of oxygen from the SNO surface. Increasing the bias voltage decrease the local near-field amplitude (local conductivity), as shown in the plot in Fig. 1i of the normalized amplitude, defined as the amplitude of the square writings ($s_{2\,\text{(writing)}}$) divided by the amplitude of the pristine SNO ($s_{2\,\text{(SNO)}}$). We note that parts of the extended weak surface potential shown on the right near the sample edge in Fig. 1c, is clearly picked up by SKPM, but it is missing in the s-SNOM images in Fig. 1d−g (see Supplementary Fig. 9

for line profile comparison). The combination of charge writing with c-AFM and optical imaging of local electronic changes by s-SNOM opens a prospect for local conductivity lithographic patterning that could be of great interest in nanophotonics.

In addition to voltage control, the local conductivity of SNO can be tuned via temperature modulation or spontaneous hydrogenation, which also induce metal-insulator transition (MIT)[35]. We first investigate the temperature dependence of the pristine and charge written areas by mapping the near-field IR local response of the film. To this end, the sample was heated in situ at different temperatures on a custom-built heating stage integrated with the s-SNOM setup. After thermal equilibrium is reached (which is achieved by keeping the system at the selected temperature for 15 min), near-field images of the sample were acquired. In Figs. 1d, e, we show two amplitude images taken at room (23 °C) and high temperature (82 °C), respectively (see Supplementary Fig. 1 for a series of images at other temperatures). Unlike most metals, for correlated oxides like SNO, increasing temperature or disorder does not hasten electron scattering processes; instead, increasing the temperature increase conductivity[40]. This is shown in Fig. 1h, which shows a linear normalized near-field amplitude plot as a function of temperature, revealing increasing amplitude, commensurate with increasing conductivity, as the temperature increases. However, the charge written areas do not show a similar trend, instead the measured change in normalized amplitude with temperature is weak (see Supplementary Fig. 2) owing to reduced oxygen content. These dissimilar changes in conductivity between the pristine and charge written areas remarkably result in making the invisible charge written at low voltage (e.g., 5 V) visible in s-SNOM

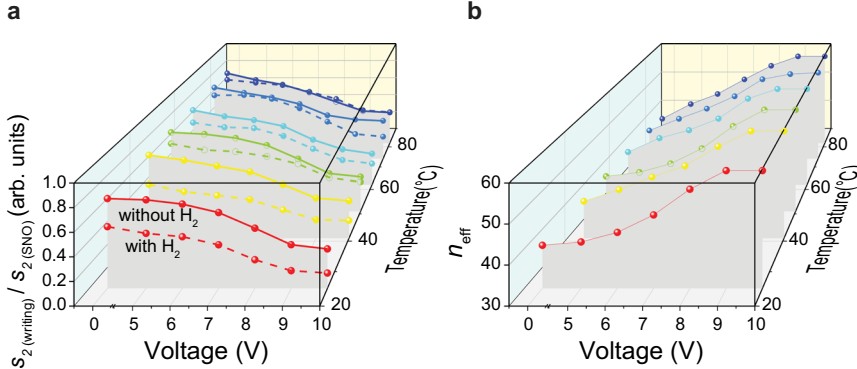

**Fig. 3 | Nanoscale control of the local conductivity of SNO. a** 3D plot showing changes in normalized amplitude $s_{2 (writing)}/s_{2 (SNO)}$ (*z*-axis) as a function voltage (*x*-axis) and temperature (*y*-axis) on the pristine and hydrogen doped SNO samples. Dashed lines represent data on a sample exposed to 5% $H_2$ and solid lines represent data on the pristine sample. **b** Ratio of polariton wavevector to free space photon wavevector vs voltage and temperature for α-MoO₃.

amplitude images, as shown in Fig. 1e. Oxygen vacancies affect the electron occupancy (donate electrons to Ni-site $e_g$ orbitals) and electron-electron correlation energy and band gap in SNO, which modulates the MIT behavior. S-SNOM images provide a direct and facile way to monitor these modulations.

Hydrogen dopants provide another robust route to dramatically modify the electronic phases of SNO[24]. We manipulate the local conductivity of pristine and charge written patterns of SNO by spontaneous hydrogen incorporation and then record in-situ the nanoscale changes via infrared s-SNOM imaging. The sample, with charge written and pristine areas, is exposed to $H_2$ using forming gas which is 5% $H_2$ and 95% $N_2$ for 30 min, while simultaneously heating the sample at 100 °C in a closed chamber. After hydrogen doping, the sample was cooled to room temperature and subsequent s-SNOM mapping was conducted as a function of temperature. Normalized amplitude s-SNOM plots of the pristine area of SNO as a function of temperature with and without hydrogen exposure are shown in Fig. 1h. The hydrogenated sample had a lower resistance than the pristine areas at all temperatures due to the modification of the electron configuration of $e_g$ orbital of Ni in SNO[27]. A change in temperature also affects the charge written areas, resulting in a large increase of conductivity at high temperatures (Supplementary Fig. 2).

**Nano-confined HPhPs hotspots on hBN/SNO**

Our technique enables facile, large scale and complex nanophotonic design of desired polaritonic structures via controllable and tunable localized conductivities of the substrate on which these polaritons propagate. This route provides a major advantage compared to complicated and nonreconfigurable conventional patterning methods such as electron beam or FIB. As an experimental demonstration, we patterned two sets of rows of cubic nanostructures with different conductivities using two different bias voltages (10 V and 8 V) of the c-AFM tip as shown in the SKPM images in Fig. 2a. Both the SKPM and s-SNOM (Fig. 2b) amplitude images of these structures faithfully reveal the larger conductivity changes in the structures written using 10 V tip potential compared to 8 V ones as expected. The s-SNOM images peak the difference showing controllable nanoscale conductivity control of nanostructures at mid IR frequencies. We then exfoliated hBN flake of thickness ~25 nm on both sets of patterns and image the nano-confined HPhP at 6.9 μm as shown in Fig. 2c. The oxygen-deficient insulating nanoscale patterns (100 nm × 100 nm) act like cavity mirrors reflecting the HPhPs at all four sides (see Supplementary Fig. 3a). Due to the large difference in permittivity between the outside and the inside of the square pattern, the polaritons are confined within the cavity and do not leak into the outside pristine region. Inside the cavity an infinite number of HPhP modes reflect from the four sides and interfere providing large field enhancement in a nano-confined volume, desirable

phenomena in optics. These series of HPhP hot-spots are shown in Fig. 2c in the s-SNOM amplitude image taken at 6.9 μm excitation wavelength. Because the square structures produced using 10 V tip-bias voltage are lower in conductivity the field intensity of the HPhP hot-spots produced are stronger by about a factor of 2 compared to the ones produced on the 8 V insulating squares (Fig. 2m). This points to an excellent route to control nano-optical field confinement and local strong nanoscale light-matter interaction. The confined mode changes in intensity profile with changing excitation wavelength further tuning the mode spatial intensity distribution as shown in the images taken at 6.45 μm (Fig. 2d) excitation wavelength (see Supplementary Fig. 4 for more wavelength and writing voltage dependence images). To further actively reconfigure these structures, we designed 8 by 7 rows of 100 nm square insulating regions on SNO and covered them with 20 nm thick hBN. Figure 2e, f shows the SKPM image and the s-SNOM amplitude image taken at 6.9 μm excitation wavelength. Similar to above results the s-SNOM images taken at taken at 6.9 μm excitation wavelength show nano-confined near-field polaritonic hotspots (Fig. 2g). We tune then the field distribution by exposing the sample to hydrogen as described above and shown in Fig. 2i, k and decrease the hotspot intensity by a factor of 2 (Fig. 2n, green color). We then further tune the intensity by erasing the writing using c-AFM negative voltage on top of the hBN and decrease the field intensity further (Fig. 2i and purple line in Fig. 2n). Nanoscale reconfigurability of the local conductivity in correlated oxides enables manipulation of sub-diffraction light-matter interactions and unique opportunities to control propagating nano-confined polariton fields. These results demonstrate a route to design a new class of cavities where multi-modal interference of HPhPs can enable tunable and reconfigurable polaritonic hotspots characterized by ultra-high field confinement and enhancement in nano-confined modal volumes.

Figure 3a summarizes the three independent knobs that enable nanoscale control of the local conductivity of SNO: oxygen vacancy control via tip voltage changes (*x*-axis), temperature (*y*-axis) and hydrogenation of the sample. Dashed lines represent normalized amplitude $s_{2(writing)}/s_{2(SNO)}$ on a sample exposed to 5% $H_2$ and solid lines represent data on pristine sample measured at various temperatures (shown by the different colors). As a case study to show the applicability of large tunability brought by the patterned SNO metasurfaces employing these knobs, we consider polaritons traveling in the [100] direction in an alpha-phase molybdenum trioxide (α-MoO₃) slab sandwiched between air and SNO, at the frequency $\omega = 990.09$ cm⁻¹. α-MoO₃ is an anisotropic van der Waals material, which has recently been explored for its unique polaritonic features[32,33,41–43]. In Fig. 3b, we plot α-MoO₃ effective index, $n_{eff}$, defined as $k_\rho/k_0$ along [100] direction, where $k_\rho$ is the in-plane momentum of PhPs and $k_0 = 2\pi/\lambda$ is the momentum of light in free space with $\lambda$ being the free-space

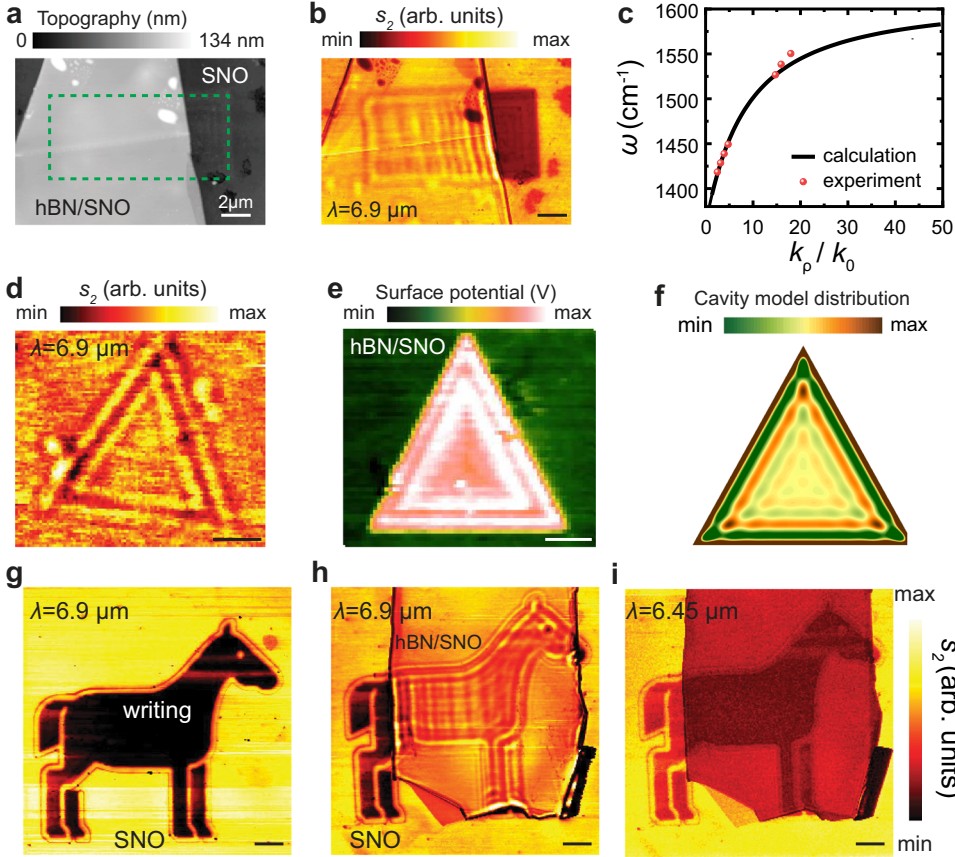

**Fig. 4 | Tunable polaritonics with hBN/SNO architectures. a** Topographic image of pristine SNO, a lithography pattern is performed by applying 10 V potential at the c-AFM tip, and a flake of 60 nm hBN (-99% boron-10 enriched) is transferred on top, green dashed lines show the charge written region on SNO. **b** IR s-SNOM second harmonic near-field amplitude $s_2$ image. **c** Measurement of the dispersion relation of HPhPs in 60 nm thick hBN on SNO. **d** Second-harmonic near-field amplitude $s_2$ image of a 60 nm hBN flake transferred on a triangular cavity lithographically patterned by applying a 10 V potential c-AFM tip. **e** Corresponding SKPM surface potential images and **f** simulation result (see "Methods") for panel (**d**). Second harmonic near-field amplitude $s_2$ images of a lithographic pattern on SNO written by applying a 10 V potential at c-AFM tip **g** prior transferring a hBN flake and **h** with a 50 nm thick hBN flake on top showing polaritons at 6.9 μm, and **i** at 6.45 μm. Scale bars indicate 2 μm.

wavelength of light. The plot shows how $n_{eff}$ can be modified as temperature, voltage, and hydrogen doping of SNO vary, which corroborates opportunities of tunable polaritonics based on correlated oxide metastructures, as further demonstrated below.

**Tunable polaritonics with hBN/SNO architectures**

We now demonstrate nanoscale hyperbolic devices based on different hBN/SNO hybrid metasurfaces by introducing spatially localized dielectric variations of patterned geometries. We used boron-10 isotopically hBN (99%)[4,30,44], a natural hyperbolic medium that supports low-loss bulk hyperbolic phonon polaritons (HPhPs)[30] exfoliated and then transferred on top of the SNO surface. Various size and shape patterns were written by applying a 5–10 V potential at the c-AFM tip. Further information on the writing process both on SNO surface as well as on hBN/SNO made by applying negative and positive potentials can be found in Supplementary Figs. 5 and 6. To demonstrate dispersion engineering, we first probe the polariton wavelength as a function of incident frequency with dielectric-like SNO, and then quantitatively extract modified HPhP dispersion. The lithography writing pattern (green dashed lines in Fig. 4a) on the surface of SNO was obtained by applying 10 V using c-AFM tip with a rectangular shape (6 μm × 12 μm); a 60 nm thick hBN (%99 enriched) was then transferred on its top covering both the writing and SNO pristine regions. We imaged HPhPs in the hBN using IR laser emitted by quantum cascade laser source focused on s-SNOM tip[45,46] (see "Methods"). Here, the AFM tip is used

to launch polaritons and collect the polaritons reflected at local domain walls between insulating and pristine phases of SNO. The evanescent fields induced at the tip apex launches HPhPs that propagate radially outward from the tip, confined within the volume of the hBN flake. Upon reaching the local pristine/charge written boundary, the HPhP is reflected and interferes with the outgoing mode to generate a pattern imaged by the s-SNOM tip as shown in Fig. 4b (a series of images are shown in Supplementary Fig. 7). To capture the dispersion of the propagating HPhPs on the SNO surface, several incident laser frequencies are exploited. Accordingly, the dispersion relation can be retrieved, as shown by red dots in Fig. 4c, which agree well with our analytical model (solid black line in Fig. 4c, see "Methods").

Next, we explore polaritonic cavity modes induced in reconfigurable hBN/SNO interfaces. Previous studies using patterned hBN such as nanoribbons[47,48] and nanocones[49] demonstrated resonant polaritons, but required demanding fabrication of insulating hBN. Here we offer an alternative facile patterning approach to realize reconfigurable polariton cavities, using spatially patterned SNO with a large contrast of conductivities. To this end, we realized a triangular cavity by applying 10 V with c-AFM tip on SNO. Figure 3d, e shows the near-field amplitude $s_2$ and surface potential SKPM images of the cavity, which agree with our numerical modeling (Fig. 4f). We model the modal polariton distribution in a cavity, in this example a triangle shape, by the interference of polaritons reflected at the edges of the substrate cavity. The detailed modeling technique is provided in the

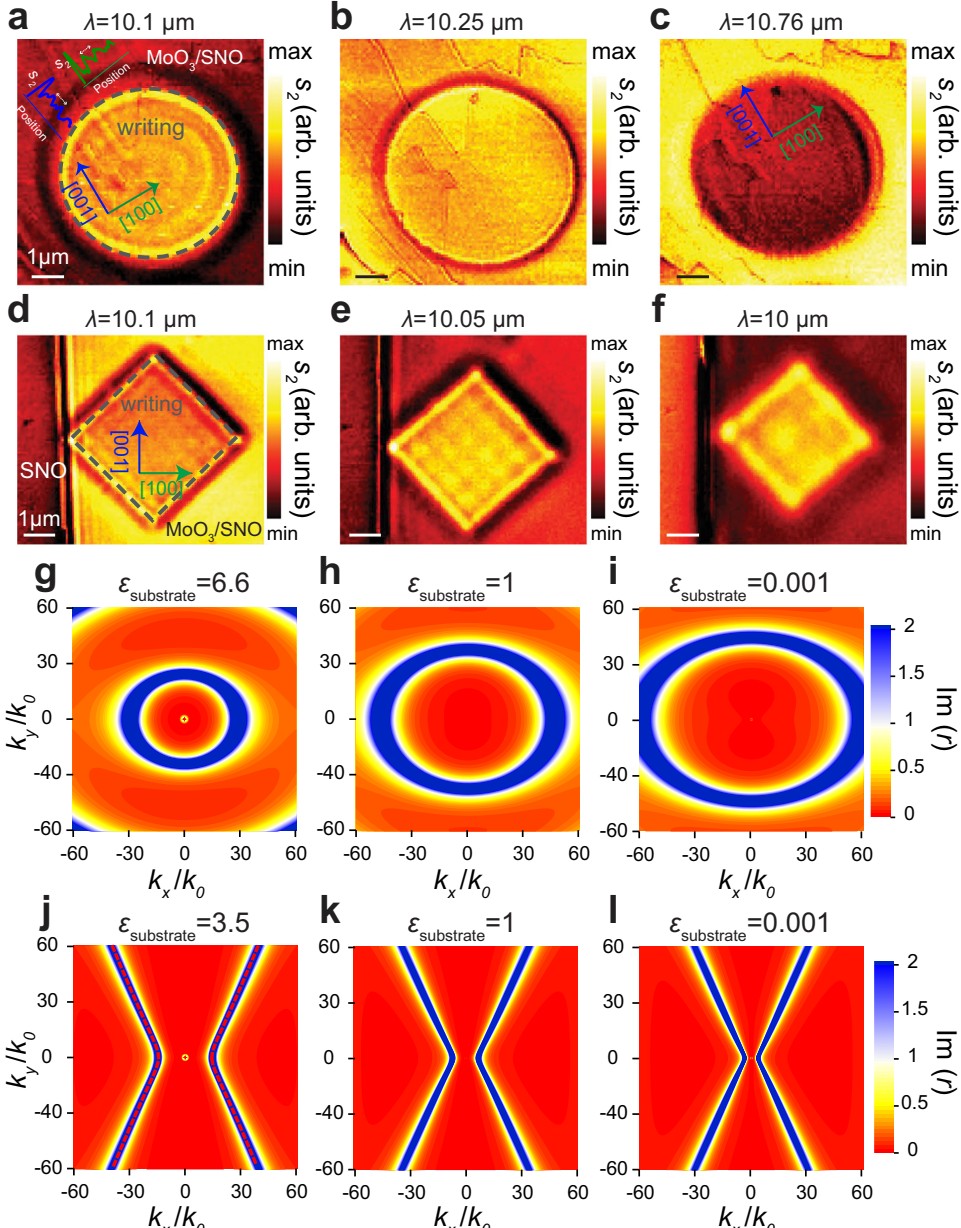

**Fig. 5 | Tunable polaritonics with α-MoO₃/SNO architectures.** IR s-SNOM second harmonic near field experimental amplitude $s_2$ images of pristine α-MoO₃/SNO at **a** $\lambda = 10.1\,\mu m$, **b** $\lambda = 10.25\,\mu m$, and **c** $\lambda = 10.76\,\mu m$. The lithography pattern (a circle with 5 μm diameter) was made by applying 10 V potential at the c-AFM tip and a flake of 120 nm α-MoO₃ is transferred on the top at parts (**a**–**c**). IR s-SNOM second harmonic near-field amplitude $s_2$ images of pristine α-MoO₃/SNO at **d** $\lambda = 10.1\,\mu m$, **e** $\lambda = 10.05\,\mu m$, **f** $\lambda = 10\,\mu m$, the lithography pattern (a square with 4 μm side) was made by applying 10 V potential at the c-AFM tip and a flake of 120 nm α-MoO₃ is transferred on the top at part (**a**–**c**). Gray dashed lines in panels **a** and **d** show the charge written region on SNO. **g**–**i** Simulated dispersion of polaritons for 100 nm thick α-MoO₃ on top of the correlated oxide metasurface with different permittivity, at the frequency of 990.09 cm⁻¹. **j**–**l** Dispersion of hyperbolic polaritons for 100 nm thick α-MoO₃ on top of correlated oxide metasurfaces with different permittivity, at the frequency of 900.9 cm⁻¹. The graphs in the insets in panel **a** represent the line profiles along the [100] and [100] directions and illustrated in detail in Supplementary Fig. 11. All scale bars in **a**–**f** indicate 1 μm.

"Methods" section. Our results also show a sharp difference for polaritons propagating into and out of the triangle cavity (Fig. 4d), further confirming the distinct properties of SNO in different phases, and hence the large reconfigurability of polaritons. The fringes of the triangular cavity originate from polariton dispersion change and not due to either non-uniform surface potentials or topography changes along the boundaries (see Supplementary Fig. 10 to see line profiles across the boundary showing lack of correlation to suggest otherwise). Moreover, different near-field images can be arbitrarily created via our modified SNO. Figure 3g, h shows a pattern written on SNO imaged at 6.9 μm laser wavelength with and without hBN flake (thickness 50 nm)

on top of SNO, demonstrating the desired metasurfaces by imaging at the polariton wavelength (Fig. 4h) or outside the range of HPhP wavelength (Fig. 4i). These examples show that a simple tip-based high-resolution patterning of the oxide surface, instead of complicated fabrication of vdW materials themselves or substrates, can provide tailorable polariton cavities and other desired designs, allowing us to tailor on-demand reconfiguration of nanoscale hyperbolic polaritons.

## Tunable polaritonics with α-MoO₃/SNO

Recent interest has been focused on in-plane anisotropic PhPs in materials such as α-MoO₃[32,33], α-V₂O₅[50], and others[51]. The dispersion of

these materials can be engineered by intercalation[50,52,53], or rotation of multiple layers[42,54]. Here, we point to another avenue to realize tunable dispersion of in-plane anisotropic PhPs via largely tunable correlated oxide metasurfaces, a reconfigurable substrate that provides nanoscale on-demand dielectric patterns enabling the active manipulation of nano-confined fields. The structures were made by c-AFM with 10 V bias, and α-MoO$_3$ was exfoliated and transferred on top of them (an anisotropic propagation of polaritons on hBN/SNO also were studied as shown in Supplementary Fig. 8). Figure 5 shows hyperbolic polaritons in α-MoO$_3$ at different wavelengths. In Fig. 5a, we observe the elliptical shape with the largest PhP wavelength along the [001] (blue plot in Fig. 5a) and the smaller value along the [100] (green plot in Fig. 5a, enlarged view of the inset is added in Supplementary Fig. 11). More importantly, as we move to the low-frequency restrstrahlen band (L-RB (around 10.76 μm) the interference patterns show an almost shape (Fig. 5c) with the longest wavelength along the [100] and almost no propagation along the orthogonal [001] direction, which agrees with recent findings in this polaritonic platform[32]. Importantly, in these images, note that HPhPs mostly propagate inside the dielectric-phase SNO, with large confinement, while the fringe periodicities are dramatic different inside and outside the SNO circles (see line profile plots in Supplementary Fig. 12). Furthermore, in Fig. 5d–f the hyperbolic modes on square patterns that are twisted 45° angle from the sample axis [001], as shown in Fig. 5d for the 10.1, 10.05, and 10 μm laser wavelength respectively. HPH reflecting from edges propagate with almost the same wavelength at L-RB, while at the twisted angle of 45° with respect to the [001] axis the anisotropy is not found, consistent with recent findings using α-MoO$_3$ nanocavities[41] and suggesting highly reconfigurable nanocavities using shaped dielectric substrates. Our correlated oxide metasurfaces unlock broad opportunities for tunable polaritonic meta-devices. To further demonstrate the great promise of their application in tunable polaritonics, we plot the analytical isofrequency polariton dispersion for 100 nm thick α-MoO$_3$ on top of correlated oxide metasurfaces with different values of permittivity, as shown in Fig. 5g–l. The large changes of momentum in different Reststrahlen bands of α-MoO$_3$ can be observed, suggesting large confinement of polaritons and different levels of light-matter interactions. Furthermore, topological transitions may also be available, if we further extend the permittivity range of tunability to negative values, as demonstrated in recent findings of α-MoO$_3$ on top of negative-permittivity substrates[55,56], which may be studied in future work.

## Discussion

In summary, nanoscale conductive regions were created on correlated prototypical perovskite SmNiO$_3$ and used to demonstrate reconfigurable hyperbolic and anisotropic phonon polariton metasurfaces. A combination of c-AFM, SKPM, and s-SNOM enabled us to generate and characterize nanoscale patterns with different conductivities values on SNO, which were further tuned via temperature modulation and spontaneous hydrogenation. Nanoscale reconfigurable conductivity control in SNO enabled manipulation of sub-diffraction light–matter interaction and dispersion engineering of desired HPhPs patterns in a direct and facile way. Our results demonstrate the potential of tunable correlated oxides metasurfaces for future configurable and tailorable quantum materials technologies.

## Methods

### SPM
Scanning probe microscopy (SPM) is a general term which includes techniques with AFM which is sharp metal probe is used in tapping mode and contacted mode. We used the Cypher AFM (oxford instrument) to acquire c-AFM and SKPM. To make the lithography pattern using c-AFM (Figs. 1, 2, and 3) a contacted mode AFM was performed while applying 5–10 V through the metal Ti/Ir coated tip

(Asyelectric.01-R2 from Oxford). In c-AFM the current is passed through the tip and into a transimpedance amplifier and it converts current to a voltage.

### SKPM
SKPM is a technique that detects the potential difference between the probe tip and the sample. This technique is based on the AC bias applied to the tip to produce an electric force on the cantilever, which is proportional to the potential difference between the tip and the sample. Using an AC bias, the probe is driven electrically and the potential difference between the tip and the sample causes the probe to oscillate. These oscillations are then canceled by a potential feedback loop and the voltage required to match the probe to the sample is recorded as the surface potential in the software.

### S-SNOM
A combination of s-SNOM and nano-FTIR is used to acquire topography, near-field images and IR nano-spectra of SNO sample prepared by PVD on LaAlO$_3$ substrates. The experimental setup (Fig. 1b, neaspec co.) is based on a tapping mode AFM with a cantilevered metal-coated tip that oscillates at a resonance frequency, Ω ~ 280 kHz and tapping amplitude of ~ 50 nm. Focused infrared laser on the metalized tip interacts with the sample, and the scattered light from this interaction is demodulated at higher harmonics nΩ of the tapping frequency and detected via phase modulation interferometer. Either a coherent broadband infrared beam in the frequency range 700–2100 cm$^{-1}$ (for nano-FTIR) or a monochromatic IR laser (for s-SNOM) is focused by a parabolic mirror to the tip. For nano-FTIR operation, the backscattered near-field light from the tip-sample junction is detected via mixing with an asymmetric Fourier transform Michelson interferometer. This detection method enables recording of both the amplitude $s(\omega)$ and phase $\varphi(\omega)$ spectra of the backscattered light. To extract background-free local near-fields, the detector signal is demodulated at a higher harmonic $n\Omega$ of the tip mechanical resonance frequency $\Omega$. Normalized amplitude ($s_n$ (sample)/$s_n$ (reference)) and phase ($\varphi_n$ (sample) -$\varphi_n$ (reference)) IR near-field spectra are acquired by first taking reference spectrum on a reference area (silicon is used in these experiments), followed by taking spectra at desired positions of the sample. (see "Methods" for details).

### Materials synthesis
**SNO**. SmNiO$_3$ thin films were prepared using magnetron co-sputtering from pure Sm and Ni target at room temperature. The substrates were cleaned using acetone and isopropanol and dried by blowing N$_2$ gas. During deposition, the chamber was maintained at 5 mtorr with flowing 40 sccm Ar and 10 sccm O$_2$ gas mixture. The sputtering power was set as 170 W (RF) for Sm and 85 W (DC) for Ni to obtain stochiometric ratio. The as deposited films were subsequently annealed at 500 °C for 24 h in high pressure oxygen gas at 1400 psi to forming the perovskite phase.

**HBN**. The hexagonal boron nitride crystal flakes were grown at atmospheric pressure using an iron-chromium solvent, isotopically-enriched boron-10 (>99% $^{10}$B), and nitrogen. The crystal growth process was previously described in detail[57].

**α-MoO$_3$**. Bulk α-MoO$_3$ crystals were synthesized via physical vapor deposition. Commercial MoO$_3$ powder (Sigma-Aldrich) was evaporated in a horizontal tube furnace at 785 °C and bulk crystals were deposited at 560 °C. The deposition was carried out in a vacuum environment, with argon as the carrier gas for vapor transport (1 Torr). Subsequently, the bulk crystals were mechanically exfoliated using adhesive tape and flakes were transferred on to 300 nm SiO$_2$ on Si substrates for characterization.

**Numerical modeling.** To model the polaritonic distribution at position $\mathbf{r}$, we take the linear combination of the polaritons launched by the tip and that reflected by the edge, which should follow[41]

$$E(\mathbf{r}) = E_0(\mathbf{r}) + \sum_m |R| e^{i\phi_R} e^{-2ik_t d_j} E_0(\mathbf{r}) \qquad (1)$$

Here, the first term is the onsite polariton signals launched by the tip, and $|R| e^{i\phi_R}$ denote the reflection characteristic at the edge by its amplitude ($|R|$) and phase parameters ($\phi_R$). The addition propagation length accounting for the polariton propagating to and reflected by the edge is included in the term $e^{-2ik_t d_j}$ where $d_j$ is the distance between the examination position and the edge and $k_t$ being in-plane complex polariton momentum. Here, for simplicity, we assume that $|R| = 1$ and the reflection phase shift is $\phi_R = 1.5\pi$, which is reasonable and gives a better fitting in main text.

## Data availability
Relevant data supporting the key findings of this study are available within the article and the Supplementary Information file. All raw data generated during the current study are available from the corresponding authors upon request.

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

## Acknowledgements

N.A.A., A.F., and Y.A. acknowledge support from the Air Force Office of Scientific Research (AFOSR), Grant No. FA9559-16-1-0172 and National Science Foundation (NSF), Grant No. 1904097. G.H. and A.A. have been supported by the Vannevar Bush Faculty Fellowship, the Office of Naval Research with grant No. N00014-22-1-2448 and the Air Force Office of Scientific Research MURI program. S.W. acknowledges project support from the Australian Research Council's (ARC) Discovery Project DP220100020. S.S. acknowledges the ARC for project support (CE200100010). S.R. acknowledges NSF DMR 1904081 for support. J.H.E. and J.L. appreciate support for hBN crystal growth from the Office of Naval Research (ONR), award no. N00014-20-1-2474.

## Author contributions

Y.A., A.A., and N.A.A. conceived and guided the experiments. N.A.A. and A.F. carried out the experiments. N.A.A. conducted the conductive writing experiments, flake transfer, and experimental data analysis. G.H. did the calculations for polariton dispersion. S.R. and Z.Z. synthesized thin films of $SmNiO_3$. J.H.E. and J.L. provided the hBN. S.B., S.W., and S.S. developed techniques for the synthesis of the α-$MoO_3$. All authors contributed to writing the manuscript.

## Competing interests

The authors declare no competing interests.
