## [Peer Review File · Nature Communications]

Reconfigurable Hyperbolic Polaritonics with Correlated Oxide MetasurfacesThis manuscript has been previously reviewed at another journal that is not operating a transparent peer review scheme. This document only contains reviewer comments and rebuttal letters for versions considered at Nature Communications.

REVIEWER COMMENTS

Reviewer #1 (Remarks to the Author):

The work discussed the modulation of phonon polaritons using a correlated oxide substrate, the SmNiO₃ (SNO). By placing α -MoO₃ and hBN, which supports in-plane anisotropic and isotropic phonon polaritons, respectively, onto the SNO, they managed to show the tailoring of the polariton wavelength as well as amplitude. These are enabled by patterning the SNO using an applied voltage, whereby the oxygen vacancies can be generated on the SNO surface and heavily modify the substrate's conductivity. They also investigated the influence of substrate temperature and hydrogenation on forming of the dielectric domain.

Tailoring of PhPs are very important for their applications in active photonic devices. In this viewpoint the work is timely and interesting. However, tuning the polaritons by engineering the dielectric properties of the supporting substrates has been extensively studied (see for example from the citations Nat. Mater. 2016, 15, 870–875.; Nat. Commun. 2018, 9, 4371.; Nat. Commun. 2019, 10, 4487.). My most concern is the novelty and advancement of their findings, because everything reported is readily expected and not too much different from previous studies using phase-change materials (Nat. Mater. 2016, 15, 870–875.; Nat. Commun. 2018, 9, 4371.) or ferroelectric materials (Nano Lett. 2015, 15, 4859–4864.) as substrates. The major difference as claimed by the author is nanoscale manipulation of phonon polaritons with the use of the SNO substrate. However, according to their results (Fig. 4a to 4f), the patterned domains, i.e., the written regions where polaritons are modified from the substrate, are still of micrometer scale. Previous studies using phase-change materials with ns laser ablation can already realize such a tunability (see for example Nat. Mater. 2016, 15, 870–875.). Most importantly, because the wavelengths of phonon polaritons are of \sim 100 nm scale, tuning of the polariton wavelength actually refers to "nanoscale tuning", and this is exactly what the previous studies have been realized. Another potential advancement by using the SNO is reversible tuning of the polaritons, because the writing region is able to be erased using a negative bias (or annealing in an oxygen-rich environment). But the authors did not make any attempts on this point.

I am sorry that I could not be positive. Here are a few technical points which I think may help improve presenting their results, especially for readers out of polaritonic community. I would strongly suggest the authors to fully addressed these points before they taking any next steps on their manuscript.

- 1) The number correspond to scale bars in Fig. 1c to 1g, Fig. 3b, 3d to 3i, and Fig. 4b to 4f should be clearly indicated.
- 2) In the abstract the authors claimed on "real-time reconfigurability". To me by using the word "real-time" usually means that the tuning should be fast and reversible. If they could not convince this point, I suggest not using the word "real-time", which could overestimate their findings.
- 3) Regarding Q2, I suggest the authors to study the reversible tuning on the phonon polaritons, for example, by using a negative bias or annealing in an oxygen-rich environment. On the basis of such study, the tuning speed can be obtained and the advantage of using the SNO strategy over the others can be reassessed.
- 4) In Fig. 1c, what is the origin of the bright region at the right edge of the sample? According to the surface potential imaging, the bright region should correspond to oxygen-vacancy rich region, which can reduce the near-field optical signal. However, in Fig. 1d to 1g, the bright region in Fig. 1c shows the same optical contrast as those in pristine SNO regions. The authors should give some explanations on this point.
- 5) In Fig. S2b, the origins of the bumps at 50 °C should be further discussed.

6) A direct quantitative correlation between Fig. 3d and Fig. 3f should be provided. Moreover, to me it looks that the writing of the triangular region can already generate fringe-like features in the surface potential distribution (Fig. 3e). Are the fringes observed in Fig. 3d originated from such non-uniform surface potentials?

7) I suggest the authors not using the word "metasurface" in their manuscript. Metasurface usually refers to artificial planar architectures with electromagnetic properties (or more general, wave properties) which cannot be realized in natural materials. To me, the structures written on the SNO surface are just some ordinary patterns. They just modify the permittivity but not bring in any designed electromagnetic functionalities.

8) Insets of Fig. 4a are hard to discern. They should be displayed with more details and more quantitatively.

9) Abbreviations should be defined when they are first used, e.g., "L-RB".

10) Data in Fig. 4 is not consistent with the related discussion in the main text. For example, the authors claimed that "...with the longest wavelength along the [100] 198 and almost no propagation along the orthogonal [001] direction,..." (Line 197 to 198 on Page 13), but I could not find any fringes in the written circle on Fig. 4c. The bright strip on the right edge should come from artifacts from the s-SNOM scanning. Another example is that the authors mentioned "...while the fringe periodicities are dramatic different inside and outside the SNO circles...." (Line 200 to 201 on Page 13), but I could not see any fringes outside of the written circle. To be more convinced on these discussions, I suggest the authors to provide quantitative line-profile analyzing on these near-field images.

11) The authors should proof-read their manuscript very carefully and correct the typos. For example, in Line 85 on Page 5, Fig. 1h should be Fig. 1c.

Reviewer #2 (Remarks to the Author):

This paper explores the manipulation of polariton optics of a 2D Van der Waals material, using the ability to change the local conductivity of an underlying oxide substrate at the nanoscale. The idea of using the underlying substrate to tailor polariton propagation is well explored in author references 6 and 7. One notion here is the resolution enabled using this method could be extremely high.

First, the authors need to more quantitatively show the change in propagation dynamics. While the strong interaction is well illustrated via s-SNOM imaging. The field induced change in polariton parameters could be better illustrated by more directly correlating the quasi-particle dispersion with the conductive nature of the SNO substrate. For example, simply measure a field induced phase delay when the polariton traverses a area with a conductive patch. PhPs are driven by polar interactions, the idea that you should see s-SNOM contrast on substrates with spatially varying conductivity is expected.

The defined resonators produced are quite interesting. Again here, the change in resonance frequency as a function of underlying field strength could be better illustrated and highlighted. This would make this manuscript stronger. The idea of defining nanoscale resonators without high-end processing is an important one. It should be discussed in more detail and rigor.

This is interesting potentially impactful work. I believe this paper could be publishable in NCOMMS, but the manuscript needs extensive revision and a little change in focus.

Reviewer #3 (Remarks to the Author):

The paper present a novel way to control polaritons in van der Waals materials using a SNO substrate which can be written by locally applying voltage with an AFM tip.

The method is very interesting due to the higher resolution compared with phase change materials, and timely. The paper is technically correct to the best of my knowledge. However there are some minor points to be addressed before acceptance

1)

There are some minor typos such as "A tip with a large positive bias remove oxygen"

2) About the sentence in the introduction:

"So far, such tunable polaritons have relied on temperature modulation, which introduces loss, as the phonon lifetime is reduced when the temperature, and correspondingly the phonon-phonon scattering events, increase. In addition, this route suffers from inaccessibility of nanoscale manipulation due to the inability to design localized thermal manipulation. As a result, reconfigurability of nanopolaritonic structures remains limited."

This sentence is misleading, since several references achieved polariton tuning without the need to maintain high temperature, and hence without impact on the polariton quality. This include ref. 7 based on GST, and the method of intercalation of MoO₃ in ref 33. The Authors should revise this sentence to clarify this point.

3)

Related to the previous point: an additional difference that should be pointed out is that phase change materials can be written and erased after the 2D materials are transferred shining light through them, while here the c-AFM tip method needs to be done prior to the transfer of the van der waals flake and is, as far as I understand, irreversible. This point should also be mentioned in the introduction to better relate this work with existing literature.

4) About the sentence:

"Unlike most metals, for correlated oxides like SNO, increasing temperature or disorder does not hasten electron scattering processes; instead, increasing the temperature increase conductivity 40"

It would be helpful to understand if this phenomenon is due to increased mobility or increased carrier's density.

5) Unless i am missing it, i cannot find the values (measured or estimated) of the permittivity / refractive index of the SNO in the two states (referred to in the supplementary as epsilon₁, and epsilon₂). These values should be provided.

6) Please clarify in the caption of figure 4 which panels are experiments and which are simulations

-----Authors Responses to Reviewers Comments are *italicised*-----
and
-----Authors addition to the manuscript and SI are **highlighted in yellow**-----

Reviewer #1 (Remarks to the Author):

The work discussed the modulation of phonon polaritons using a correlated oxide substrate, the SmNiO₃ (SNO). By placing α -MoO₃ and hBN, which supports in-plane anisotropic and isotropic phonon polaritons, respectively, onto the SNO, they managed to show the tailoring of the polariton wavelength as well as amplitude. These are enabled by patterning the SNO using an applied voltage, whereby the oxygen vacancies can be generated on the SNO surface and heavily modify the substrate's conductivity. They also investigated the influence of substrate temperature and hydrogenation on forming of the dielectric domain.

Tailoring of PhPs are very important for their applications in active photonic devices. In this viewpoint the work is timely and interesting.

Author Response #1:

We thank the referee for their valuable questions and positive comments. We understand that the main questions/comments are related to the novelty and advancement of our findings. To address these concerns and questions, we conducted several significant additional experiments. These have been included in a thoroughly revised manuscript and supplemental information. We believe that these additions have improved our manuscript considerably. Below, we detail the changes and additions we have made as well as provide a detailed response to each of the reviewer's questions.

However, tuning the polaritons by engineering the dielectric properties of the supporting substrates has been extensively studied (see for example from the citations Nat. Mater. 2016, 15, 870–875.; Nat. Commun. 2018, 9, 4371.; Nat. Commun. 2019, 10, 4487.). My most concern is the novelty and advancement of their findings, because everything reported is readily expected and not too much different from previous studies using phase-change materials (Nat. Mater. 2016, 15, 870–875.; Nat. Commun. 2018, 9, 4371.) or ferroelectric materials (Nano Lett. 2015, 15, 4859–4864.) as substrates. The major difference as claimed by the author is nanoscale manipulation of phonon polaritons with the use of the SNO substrate. However, according to their results (Fig. 4a to 4f), the patterned domains, i.e., the written regions where polaritons are modified from the substrate, are still of micrometer scale. Previous studies using phase-change materials with ns laser ablation can already realize such a tunability (see for example Nat. Mater. 2016, 15, 870–875.). Most importantly, because the wavelengths of phonon polaritons are of ~100 nm scale, tuning of the polariton wavelength actually refers to “nanoscale tuning”, and this is exactly what the previous studies have been realized. Another potential advancement by using the SNO is reversible tuning of the polaritons, because the writing region is able to be erased using a negative bias (or annealing in an oxygen-rich environment). But the authors did not make any attempts on this point.

Author Response #2:

We thank the reviewer for these questions and concerns as they have enabled us to extensively improve the manuscript. We have performed new experiments and analysis and included these in the main text of the manuscript and the SI. Below we describe these additional experiments and analysis and indicate the differences from previous works the reviewer indicated.

The major novelty and advancement of our technique are:

(i) nanoscale controllable, scalable manipulation of phonon polaritons with the use of the SNO substrate and (ii) reversible tuning, volatile as well as non-volatile switchable dielectric environment.

(i) nanoscale controllable, scalable manipulation of phonon polaritons with the use of the SNO substrate

We have performed new experiments to demonstrate the capability and potential of our technique for facile nanophotonic design of polaritonic structures that can potentially be scalable and complex design can be implemented. These results are now included as a new figure (Fig. 2) in the manuscript. Patterns of two sets of rows of squares (each square 100 nm x 100 nm) of nanostructures with different conductivities created by using c-AFM bias voltages (10 V and 8V) are shown in the SKPM and s-SNOM amplitude images (Fig. 2a and Fig. 2b, respectively). We then exfoliated hBN flake of thickness ~20 nm on both sets of patterns and image the nano-confined HPhP at 6.9 micron. The oxygen deficient insulating nanoscale patterns (each nano-square) act like cavity mirrors reflecting the HPhPs at all four sides. Due to the large difference in permittivity between the outside and the inside of the square pattern, the polaritons are confined within the cavity and do not leak into the outside SNO region. Inside the cavity an infinite number of HPhP modes reflect from the four sides and interfere, providing large field enhancement in a nano-confined volume, extremely desirable phenomena in optics. These series of wavelength dependent HPhP hot-spots are shown in Fig. 2c,d in the s-SNOM amplitude image taken at 6.9 μm and 6.45 μm excitations. Because the square structures produced using 10 V tip-bias voltage are lower in conductivity, the field intensity of the HPhP hot-spots produced are stronger by about a factor of 2 compared to the ones produced on the 8 V insulating squares (Fig. 2m). This points to an excellent route to control nano-optical field confinement and local strong nanoscale light-matter interaction distinct from any of the methods in References (Nat. Mater. 2016, 15, 870–875 Taubner, Nat. Comm. 2018, 9, 4371-VO2 our work, Nat. Comm. 2019, 10, 4487-Capasso). In both Ref (Nat. Mater. 2016, 15, 870–875 Taubner and Nat. Commun. 2019, 10, 4487) the size of the structures is diffraction limited whereas earlier work reported in Ref (Nat. Comm. 2018, 9, 4371) is limited since VO₂'s different phases cannot co-exist in a controllable manner due to thermal nature of the transition. Our results demonstrate a novel route to design a new class of cavities where multimodal interference of HPhPs can enable tunable and reconfigurable polaritonic hotspots characterized by ultra-high field confinement and enhancement in nano-confined modal volumes.

To highlight these results, we have added extended discussion in the manuscript and modified Figure 2.

(ii) reversible tuning, volatile as well as non-volatile switchable dielectric environment.

To actively reconfigure these structures and demonstrate reversible tuning of our design we performed additional experiments and included the results also in Fig. 2 as well as in SI which we describe below.

As shown in Fig. 2e-l we designed 8 by 7 rows of 100 nm square insulating regions on SNO and covered them with ~20 nm thick hBN. Similar to above results the s-SNOM images taken at 6.9 μm excitation wavelength show (Fig. 2g) nano-confined near-field polaritonic hotspots and wavelength dependence intensity profile (Fig. 2h). We then tune the field distribution of the patterns (Fig. 2i shows the patterns with the Pt electrode) by exposing the sample to hydrogen as shown in Fig. 2j. This decreases the hotspot intensity by a factor of 2 (Fig. 2n green color). We then further tune the intensity by erasing the writing using c-AFM negative voltage on top of the hBN and decrease the field intensity further (Fig. 2l and purple line in Fig 2n).

The new discussion included in the manuscript and modified Figure 2 included are shown below.

Fig. 2 | Nanophotonic design of HPhPs structures on hBN via controllable and tunable localized conductivities of the NNO substrate. a, SKPM surface potential image of pristine SNO with conductive writing patterns made by applying potential at the c-AFM tip, 10 V (three top rows) and 8 V (three bottom rows). b, Corresponding s-SNOM second harmonic near-field amplitude, s_2 , image of panel 'a' taken at $\lambda=6.9 \mu\text{m}$ excitation wavelength. S-SNOM second harmonic near-field amplitude, s_2 , image of panel 'a' with a hBN flake transferred on top of the pattern imaged at $\lambda=6.9 \mu\text{m}$ excitation wavelength (c) and at $\lambda=6.45 \mu\text{m}$ (d). e, SKPM surface potential image of pristine SNO with conductive writing patterns made by applying 10 V potential at the c-AFM

tip. **f**, corresponding s-SNOM second harmonic near-field amplitude, s_2 , image of panel 'e' taken at $\lambda=6.9 \mu\text{m}$ excitation wavelength. S-SNOM second harmonic near-field amplitude, s_2 , image of panel 'a' with a hBN flake transferred on top of the pattern imaged at $\lambda=6.9 \mu\text{m}$ excitation wavelength (**g**) and at $\lambda=6.45 \mu\text{m}$ (**h**). **i**, near-field second harmonic amplitude, s_2 , images of the conductive pattern on hBN/ SNO and Pt electrode shown on the left of the patterns before H_2 exposure (**i**), after H_2 exposure (**j**), zoomed in image after H_2 exposure (**k**) followed by erasing made by applying 10V potential at the c-AFM tip (**l**). **m**, near-field second harmonic amplitude, s_2 , line profiles extended along the dash lines in panels c, d, e and d (blue, orange, green and violet, respectively). **n**, near-field second harmonic amplitude, s_2 , line profiles extended along the dash lines in panels j, k, and l (orange, neon green and violet respectively). **o**, 3D plot showing changes in normalized amplitude $S_{2(\text{writing})}/S_{2(\text{SNO})}$ (z-axis) as a function voltage (x-axis) and temperature (y-axis) on the pristine and hydrogen doped SNO samples. Dashed lines represent data on a sample exposed to 5% H_2 and solid lines represent data on the pristine sample. **p**, Ratio of polariton wavevector to free space photon wavevector vs voltage and temperature. All scale bars indicate $1 \mu\text{m}$.

Our technique enables facile, and complex nanophotonic design of desired polaritonic structures via controllable and tunable localized conductivities of the substrate on which these polaritons propagate. This route provides an advantage compared to complicated and static conventional patterning methods such as electron beam or FIB. As an experimental demonstration we patterned two sets of rows of cubic nanostructures with different conductivities using two different bias voltages (10 V and 8V) of the c-AFM tip as shown in the SKPM images in Fig. 2a. Both the SKPM and s-SNOM (Fig. 2b) amplitude images of these structures faithfully reveal the larger conductivity changes in the structures written using 10 V tip potential compared to 8 V ones as expected. The s-SNOM images peak the difference showing controllable nanoscale conductivity control of nanostructures at mid IR frequencies. We then exfoliated hBN flake of thickness $\sim 20 \text{ nm}$ on both sets of patterns and image the nano-confined HPhP at 6.9 micron as shown in Fig. 2c. The oxygen deficient insulating nanoscale patterns ($100 \text{ nm} \times 100 \text{ nm}$) act like cavity mirrors reflecting the HPhPs at all four sides. Due to the large difference in permittivity between the outside and the inside of the square pattern, the polaritons are confined within the cavity and do not leak into the outside pristine region. Inside the cavity an infinite number of HPhP modes reflect from the four sides and interfere,

providing large field enhancement in a nano-confined volume, extremely desirable phenomena in optics. These series of HPhP hot-spots are shown in Fig. 2c in the s-SNOM amplitude image taken at 6.9 μm excitation wavelength. Because the square structures produced using 10 V tip-bias voltage are lower in conductivity the field intensity of the HPhP hot-spots produced are stronger by about a factor of 2 compared to the ones produced on the 8 V insulating squares (Fig. 2m). This points to a promising route to control nano-optical field confinement and local strong nanoscale light-matter interaction. The confined mode changes in intensity profile with changing excitation wavelength further tuning the mode spatial intensity distribution as shown in the images taken at 6.45 μm excitation wavelength (Fig. 2d). To further actively reconfigure these structures, we designed 8 by 7 rows of 100 nm square insulating regions on SNO and covered them with 20nm thick hBN. Figure 2e and 2f show the SKPM image and the s-SNOM amplitude image taken at 6.9 μm excitation wavelength. Similar to above results the s-SNOM images taken at taken at 6.9 μm excitation wavelength show nano-confined near-field polaritonic hotspots (Fig. 2g). We tune then the field distribution by exposing the sample to hydrogen as described above and show in Fig. 2i and 2k and decrease the hotspot intensity by a factor of 2 (Fig. 2n green color). We then further tune the intensity by erasing the writing using c-AFM negative voltage on top of the hBN and decrease the field intensity further (Fig. 2i and purple line in Fig 2n). Nanoscale reconfigurability of the local conductivity in correlated oxides enables manipulation of sub-diffraction light-matter interactions and unique opportunities to control propagating nano-confined polariton fields. These results demonstrate a novel route to design a new class of cavities where multimodal interference of HPhPs can enable tunable and reconfigurable polaritonic hotspots characterized by ultra-high field confinement and enhancement in nano-confined modal volumes.

Further Author Response:

An important advantage of using the SNO platform for nanophotonics is the ability to reversibly tune the pattern surface thereby tuning polaritons via writing with a positive tip bias and erasing a region using a negative bias. This write/erase mode can be applied either to the bare SNO surface or to the hBN/SNO surface enabling interesting possibilities for nanophotonics. To that end we wrote a square pattern on pristine SNO by applying 8V potential at the c-AFM tip (see Fig. S4 a, Topography b, SKPM surface potential image). We then exfoliated a 45 nm thick hBN, followed by erasing several circular patterns by applying -10V potential at the c-AFM tip (cyan dash boxes, Fig. S4 e) and -8V potential at the c-AFM tip

(cyan dash circle, Fig. S4 f) on top of hBN/SNO. This demonstrates proof-of-concept write/erase ability on the heterostructure hBN/SNO platform. In Fig. S5 we wrote all patterns on top of pristine hBN/SNO and erased also on top of hBN/SNO. We also note that these process of writing/erasing on top of hBN does not seem to damage the hBN surface judging from the topographic profile we show in Fig. S5g perhaps due to its hBN is a dielectric with a high breakdown field. However, the effect of writing on hBN need further investigation¹.

We have added the following figures in the SI.

Fig. S5 | **a**, Topography of hBN on pristine SNO, **b**, SKPM surface potential image of conductive patterns written on top of hBN/SNO made by applying 6V potential to the c-AFM tip (cyan dash box) **c**, SKPM surface potential image made by applying -6V potential (red dash box) on the same region as **b** (effectively erasing of potential shown in **b**). **d**, Larger topography image of the same flake shown in **a**, **e**, SKPM surface potential image of a conductive larger square patterns written on top of hBN/SNO made by applying 8V potential at the c-AFM tip (cyan dash box) **f**, SKPM image of the same region as **e** but a concentric erasing shown in the middle made by applying -8V potential at the c-AFM tip (red dash box). **g**, topography line profile taken on **d** (shown by the green broken line) also shown in **f**. No meaningful topographic change (within the z-axis resolution of our AFM) has been observed on hBN surface.

I am sorry that I could not be positive. Here are a few technical points which I think may help improve presenting their results, especially for readers out of polaritonic community. I would strongly suggest the

authors to fully address these points before they taking any next steps on their manuscript.

Author Response #3:

On the contrary we very much appreciate all the important comments and questions. These were great motivation for us to add important new results that certainly improved the manuscript. Without the Reviewer's advice we probably would not have done these new measurements.

1) The number correspond to scale bars in Fig. 1c to 1g, Fig. 3b, 3d to 3i, and Fig. 4b to 4f should be clearly indicated.

Author Response #4:

Done.

2) In the abstract the authors claimed on “real-time reconfigurability”. To me by using the word “real-time” usually means that the tuning should be fast and reversible. If they could not convince this point, I suggest not using the word “real-time”, which could overestimate their findings.

Author Response #5:

We thank the reviewer for this comment. In Fig. 2i, j, n, o & p we believe we have demonstrated real-time tuning of the polariton dispersion as well as the hotspot field distributions by exposing the patterned hBN/SNO heterostructure with hydrogen (Fig. 2i, j). This process decreased the hotspot intensity by a factor of 2 (Fig. 2n green color). As the hydrogen diffuses out of SNO, the SNO as well as the polariton resonator hotspots recover their initial state over time. This is clearly represented in Fi. S3b. The black points (black line) show the s-SNOM contrast recovering from an insulating state to its initial pristine state as time increases after doping. We also measure the amplitude contrast at 3 points on the square designs we created, specifically in Fig. 2g. The pink line plot represents the s-SNOM amplitude contrast measured on a polariton hotspot normalized with the Pt surface as shown on the right on the y-axis in Fig. S3. Before doping the structures with hydrogen, the hotspots have a contrast of ~ 0.33 shown by the broken pink horizontal lines. As we dope with hydrogen the hotspots become more pristine (~ 0.45) because the SNO is becoming more insulating as revealed by the black points. As time increases hydrogen diffuses out slowly from under hBN so SNO becomes more pristine thereby decreasing the hotspots contrast (making them less metallic) as shown by the decreasing trend of both the pink and the blue lines. Although the time scale in our experiment is slow, it can be hastened by accelerating the hydrogen diffusion for example by heating the sample or applying electric fields to re-distribute the hydrogen dopants. The method may enable real-time tunability, but we do not claim in this work that the reversibility is fast.

In order to make sure we do not overstate our results, following the Reviewer's comment, we have removed the word “real-time” from the suggested line in the abstract

3) Regarding Q2, I suggest the authors to study the reversible tuning on the phonon polaritons, for example, by using a negative bias or annealing in an oxygen-rich environment. On the basis of such study, the tuning speed can be obtained and the advantage of using the SNO strategy over the others can be reassessed.

Author Response #6:

We thank the reviewer for this suggestion. We have performed these experiments and described our findings as discussed above in our response to reviewer's question above (Author Response#2).

4) In Fig. 1c, what is the origin of the bright region at the right edge of the sample? According to the surface potential imaging, the bright region should correspond to oxygen-vacancy rich region, which can reduce the near-field optical signal. However, in Fig. 1d to 1g, the bright region in Fig. 1c shows the same optical contrast as those in pristine SNO regions. The authors should give some explanations on this point.

Author Response #7:

We appreciate the Reviewer’s careful observation. The difference in sensitivity between the two techniques resulted in the observed difference between the SKPM and s-SNOM images. In general, we have repeatedly found out that SKPM is more sensitive in detecting small changes in conductivity than IR s-SNOM. The strong insulating phase near the edge of the Pt is clearly picked up by s-SNOM (shown in dark contrast near the Pt edge on s-SNOM images). Parts of the extended weak surface potential, which is clearly picked up by SKPM, is missing/very weak on s-SNOM images due to its reduced sensitivity compared to SKPM. To indicate these differences for the reader, we have added the following text on page 6 top paragraph:

We note that parts of the extended weak surface potential shown on the right near the sample edge in Fig. 1c, is clearly picked up by SKPM but is missing in the s-SNOM images in Fig. 1d-g (see Fig. S8 for line profile comparison).

We have also added a line profile comparison of SKPM and s-SNOM images in the SI (Fig. S8) shown below:

extended insulating region. (c) Show the line profiles taken at the SKPM image (orange line) and the near-field image (blue line). Not all of the extended insulating region shown in the SKPM is picked up by the s-SNOM due to difference in sensitivity.

5) In Fig. S2b, the origins of the bumps at 50 °C should be further discussed.

Author Response #8:

We appreciate the Reviewer's careful observation. This bump is caused by the non-uniformity of the written squares. Each normalized amplitude points in Fig. S2b plot were extracted by averaging 3x3 pixels of the s-SNOM signal at the middle of each square pattern followed by normalization with signal on Pt electrode. Nonuniformly oxygen depleted region temperature dependence response could lead to fluctuations (bumps) such as observed in Fig. S2b.

To clarify this point to the reader we have added the following text on the SI on the caption of Fig. S2:

Each normalized amplitude points on these plots were extracted by averaging 3x3 pixels of the s-SNOM signal at the middle of each square pattern followed by normalization with signal on Pt electrode. The bumps shown in (b) at 50 °C could be caused due to the written squares not being uniformly insulating affecting the temperature dependent s-SNOM signal.

6) A direct quantitative correlation between Fig. 3d and Fig. 3f should be provided. Moreover, to me it looks that the writing of the triangular region can already generate fringe-like features in the surface potential distribution (Fig. 3e). Are the fringes observed in Fig. 3d originated from such non-uniform surface potentials?

Author Response #9:

We thank the reviewer for this question. We have shown with even smaller cavity design in Fig. 2 that the fringes originate from polariton dispersion change and not due to either non-uniform surface potentials or topography changes along the boundaries. To clarify this point and provide comparison we have added a figure in the SI, Fig. S9 that shows lack of strong correlation between nano-uniformity of the topography and surface potential changes with the observed fringes.

We have added the following sentence on the main manuscript on page 14 (top paragraph) to indicate this point:

The fringes of the triangular cavity originate from polariton dispersion change and not due to either non-uniform surface potentials or topography changes along the boundaries (see Fig. S9 to see line profiles across the boundary showing lack of correlation to suggest otherwise).

We have also added the following figure in the SI.

7) I suggest the authors not using the word “metasurface” in their manuscript. Metasurface usually refers to artificial planar architectures with electromagnetic properties (or more general, wave properties) which cannot be realized in natural materials. To me, the structures written on the SNO surface are just some ordinary patterns. They just modify the permittivity but not bring in any designed electromagnetic functionalities.

Author Response #10:

We thank the reviewer for this suggestion. This is the only point on which we respectfully disagree with the Reviewer. With the addition of Fig. 2 we believe the written structures do bring in designed electromagnetic functionalities. Our results in Fig. 2 represent a novel design of a new class of cavities where multimodal interference of HPhPs can enable tunable and reconfigurable polaritonic hotspots characterized by ultra-high field confinement and enhancement in nano-confined modal volumes. We believe this is consistent with the terminology of metasurfaces described in literature.

8) Insets of Fig. 4a are hard to discern. They should be displayed with more details and more quantitatively.

Author Response #11:

We thank the reviewer for this suggestion. We have added a SI figure where we show the line profiles clearly for different axis.

To describe these changes we added the following figures in the SI and text in the main manuscript:

On page 16 of the manuscript:

“..enlarged view of the inset is added in Fig. S10”

The following figure in the SI:

Fig. S10 | Enlarged image of the inset in Fig. 4a of the main manuscript. **a**, line profile along the [100] in Fig. 4a showing the polariton propagation perpendicular to the edge of α -MoO₃ flake. **b**, line profile along the [001] in Fig. 4a showing the polariton propagation parallel to the edge of α -MoO₃ flake.

9) Abbreviations should be defined when they are first used, e.g., “L-RB”.

Author Response #12:

We thank the reviewer for this comment. Here, “L-RB” means the low-frequency Reststrahlen Band (L-RB). We have added these changes on page 16 of the manuscript. We also checked the whole manuscript to make sure the definition of abbreviation is defined when first used, including c-AFM (conductive atomic force microscopy), s-SNOM (scattering-type scanning near-field optical microscopy) and others.

These changes are indicated on page 5 of the manuscript.

10) Data in Fig. 4 is not consistent with the related discussion in the main text. For example, the authors claimed that “...with the longest wavelength along the [100] 198 and almost no propagation along the orthogonal [001] direction...” (Line 197 to 198 on Page 13), but I could not find any fringes in the written circle on Fig. 4c. The bright strip on the right edge should come from artifacts from the s-SNOM scanning. Another example is that the authors mentioned “...while the fringe periodicities are dramatic different inside and outside the SNO circles...” (Line 200 to 201 on Page 13), but I could not see any fringes outside of the written circle. To be more convinced on these discussions, I suggest the authors to provide quantitative line-profile analyzing on these near-field images.

Author Response #13:

We thank the Reviewer for this suggestion. We have provided line profiles of the fringes across two axes and as a function of wavelength for images shown in Fig. 4a-c. Similar to what we show in Fig. S10 the line profile along the [100] shows a shorter wavelength of the polariton compared with the one along the [100] (polariton propagation parallel to the edge of α -MoO₃ flake). A similar, although weak, axis dependence of the polariton wavelength is also plotted at 10.76 μm (orange line in d and light blue line in e). The line profiles in either axes in Fig 4b (Fig. S11 b) are flat due to the absence of polariton at 10.25 μm (green lines in d and e).

To highlight these additions, we have indicated the addition of Fig. S11 on page 16 of the manuscript added the following figure in the SI, Fig. S11.

Fig. S11 | Line-profiles on the near-field images in Fig. 4a-c. of the manuscript, (a-c) show the same near-field images as Fig. 4a-c but with broken lines added along the [100] and [001] axis. d. line profiles along the [001] and e along [100] axis in Fig. 4a-c. Like in Fig. S10 the line profile along the [100] shows a shorter wavelength of the polariton compared with the one along the [100] (polariton propagation parallel to the edge of α -MoO₃ flake). A similar, although weak, axis dependence of the polariton wavelength is also plotted at 10.76 μm (orange line in d and light blue line in e). The line profiles in either axis in Fig 4b (Fig. S11 b) are flat due to the absence of polariton at 10.25 μm (green lines in d and e).

11) The authors should proof-read their manuscript very carefully and correct the typos. For example, in Line 85 on Page 5, Fig. 1h should be Fig. 1c.

Author Response #14:

We thank the Reviewer for this correction. We have carefully read and made grammar and typo corrections throughout the manuscript including changing Fig. 1h to Fig. 1c on page 5.

Reviewer #2 (Remarks to the Author):

This paper explores the manipulation of polariton optics of a 2D Van der Waals material, using the ability to change the local conductivity of an underlying oxide substrate at the nanoscale. The idea of using the underlying substrate to tailor polariton propagation is well explored in author references 6 and 7. One notion here is the resolution enabled using this method could be extremely high.

Author Response #1:

We thank the Referee for recognizing the value of our work and very valuable questions/suggestions and positive comments.

First, the authors need to more quantitatively show the change in propagation dynamics. While the strong interaction is well illustrated via s-SNOM imaging. The field induced change in polariton parameters could be better illustrated by more directly correlating the quasi-particle dispersion with the conductive nature of the SNO substrate. For example, simply measure a field induced phase delay when the polariton traverses a area with a conductive patch. PhPs are driven by polar interactions, the idea that you should see s-SNOM contrast on substrates with spatially varying conductivity is expected.

Author Response #2:

*We thank the Reviewer for this comment. To address this issue sufficiently we have performed new experiments and included extensive discussion and a new figure (Fig. 2) in the manuscript. We have created nano-resonators on different conductivity level of SNO and manipulated the polariton dynamics further within the resonators by hydrogen doping and write/erase procedure. Elaboration of these structures and the propagation dynamics is given below in our response to Reviewer 1 question (see above **Author Response #1**).*

Here we also provide a summary of the new results on the quasi-particle dispersion and its very interesting changes with the changing conductive nature of the SNO substrate. We created patterns of two sets of rows of squares (each square 100 nm x 100 nm) of nanostructures with different conductivities by using c-AFM bias voltages (10 V and 8V) are shown in the SKPM and s-SNOM amplitude images (Fig. 2a and Fig. 2b respectively). We then exfoliated hBN flake of thickness ~20 nm on both sets of patterns and image the nano-confined HPhP at 6.9 micron. The oxygen deficient insulating nanoscale patterns (each nano-square) act like cavity mirrors reflecting the HPhPs at all four sides. Due to the large difference in permittivity between the outside and the inside of the square pattern, the polaritons are confined within the cavity and do not leak into the outside pristine SNO region. Inside the cavity an infinite number of HPhP modes reflect from the four sides and interfere providing large field enhancement in a nano-confined volume. These series of wavelength dependent HPhP hot-spots are shown in Fig. 2c,d in the s-SNOM amplitude image taken at 6.9 μm and 6.45 μm excitations. Because the square structures produced using 10 V tip-bias voltage are lower in conductivity, the field intensity of the HPhP hot-spots produced are stronger by about a factor of 2 compared to the ones produced on the 8 V insulating squares (Fig. 2m). These results demonstrate a novel route to design new types of nano-resonators that are tunable, with highly confined and enhanced field. We tune the field distribution of the patterns (Fig. 2i shows the patterns with the Pt electrode) by exposing the sample to hydrogen as shown in Fig. 2j. This decreases the hotspot intensity by a factor of 2 (Fig. 2n green color). We then further tune the intensity by erasing the writing using c-AFM negative voltage on top of the hBN and decrease the field intensity further (Fig. 2l and purple line in Fig 2n). These

results are shown in the line-profile plots in Fig. 2m and 2n (further discussion is given below in **Author Response #3**).

The defined resonators produced are quite interesting. Again here, the change in resonance frequency as a function of underlying field strength could be better illustrated and highlighted. This would make this manuscript stronger. The idea of defining nanoscale resonators without high-end processing is an important one. It should be discussed in more detail and rigor.

Author Response #3:

We thank the Reviewer for this excellent suggestion. We have performed new experiments and demonstrated nano-resonators and included these results as a new figure (Fig. 2) and extensive description on pages 8-10 of the manuscript. In addition, we also gave highlights of these results in our response to Reviewer 1 above (Author Response #2). We would like to add that this Reviewer's comment and suggestion played an important role in enhancing our work: (1) it helped us think about decreasing the resonator dimension to truly nanoscale (as we show in the new figure we added in the revised manuscript, Fig. 2) compared to our original structures shown in Fig. 3d-f; 2) we could easily visualize and elaborate "the field induced change in polariton parameters could be better illustrated by more directly correlating the quasi-particle dispersion with the conductive nature of the SNO substrate" in the newly created nano-resonators.

*As we described above in our response to Reviewer 1 (Author Response #1) as well as in **Author Response #2:** of the Reviewer's question, we have added the following figure and accompanying text in the manuscript:*

Fig. 2 | Nanophotonic design of HPhPs structures on hBN via controllable and tunable localized conductivities of the NNO substrate. **a**, SKPM surface potential image of pristine SNO with conductive writing patterns made by applying potential at the c-AFM tip, 10 V (three top rows) and 8 V (three bottom rows). **b**, corresponding s-SNOM second harmonic near-field amplitude, s_2 , image of panel 'a' taken at $\lambda=6.9 \mu\text{m}$ excitation wavelength. S-SNOM second harmonic near-field amplitude, s_2 , image of panel 'a' with a hBN flake transferred on top of the pattern imaged at $\lambda=6.9 \mu\text{m}$ excitation wavelength (c) and at $\lambda=6.45 \mu\text{m}$ (d). **e**, SKPM surface potential image of pristine SNO with conductive writing patterns made by applying 10 V potential at the c-AFM tip. **f**, corresponding s-SNOM

second harmonic near-field amplitude, s_2 , image of panel 'e' taken at $\lambda = 6.9 \mu\text{m}$ excitation wavelength. **S**-SNOM second harmonic near-field amplitude, s_2 , image of panel 'a' with a hBN flake transferred on top of the pattern imaged at $\lambda = 6.9 \mu\text{m}$ excitation wavelength (g) and at $\lambda = 6.45 \mu\text{m}$ (h). **i**, near-field second harmonic amplitude, s_2 , images of the conductive pattern on hBN/ SNO and Pt electrode shown on the left of the patterns before H_2 exposure (**i**), after H_2 exposure (j), zoomed in image after H_2 exposure (k) followed by erasing made by applying 10V potential at the c-AFM tip (l). **m**, near-field second harmonic amplitude, s_2 , line profiles extended along the dash lines in panels c, d, e and d (blue, orange, green and violet, respectively). **n**, near-field second harmonic amplitude, s_2 , line profiles extended along the dash lines in panels j, k, and l (orange, neon green and violet respectively). **o**, 3D plot showing changes in normalized amplitude $s_{2(\text{writing})}/s_{2(\text{SNO})}$ (z-axis) as a function voltage (x-axis) and temperature (y-axis) on the pristine and hydrogen doped SNO samples. Dashed lines represent data on a sample exposed to 5% H_2 and solid lines represent data on the pristine sample. **p**, Ratio of polariton wavevector to free space photon wavevector vs voltage and temperature. All scale bars indicate $1 \mu\text{m}$.

Our technique enables facile and complex nanophotonic design of desired polaritonic structures via controllable and tunable localized conductivities of the substrate on which these polaritons propagate. This route provides a huge advantage compared to complicated and nonreconfigurable conventional patterning methods such as electron beam or FIB. As an experimental demonstration we patterned two sets of rows of cubic nanostructures with different conductivities using two different bias voltages (10 V and 8V) of the c-AFM tip as shown in the SKPM images in Fig. 2a. Both the SKPM and s-SNOM (Fig. 2b) amplitude images of these structures faithfully reveal the larger conductivity changes in the structures written using 10 V tip potential compared to 8 V ones as expected. The s-SNOM images peak the difference showing controllable nanoscale conductivity control of nanostructures at mid IR frequencies. We then exfoliated hBN flake of thickness $\sim 25 \text{ nm}$ on both sets of patterns and image the nano-confined HPhP at 6.9 micron as shown in Fig. 2c. The oxygen deficient insulating nanoscale patterns (100 nm x 100 nm) act like cavity mirrors reflecting the HPhPs at all four sides. Due to the large difference in permittivity between the outside and the inside of the square pattern, the polaritons are confined within the cavity and do not leak into the outside pristine region. Inside the cavity an infinite number of HPhP modes reflect from the four sides and

interfere providing large field enhancement in a nano-confined volume, extremely desirable phenomena in optics. These series of HPhP hot-spots are shown in Fig. 2c in the s-SNOM amplitude image taken at 6.9 μm excitation wavelength. Because the square structures produced using 10 V tip-bias voltage are lower in conductivity the field intensity of the HPhP hot-spots produced are stronger by about a factor of 2 compared to the ones produced on the 8 V insulating squares (Fig. 2m). This points to an excellent route to control nano-optical field confinement and local strong nanoscale light-matter interaction. The confined mode changes in intensity profile with changing excitation wavelength further tuning the mode spatial intensity distribution as shown in the images taken at 6.45 μm excitation wavelength (Fig. 2d). To further actively reconfigure these structures, we designed 8 by 7 rows of 100 nm square insulating regions on SNO and covered them with 20nm thick hBN. Figure 2e and 2f show the SKPM image and the s-SNOM amplitude image taken at 6.9 μm excitation wavelength. Similar to above results the s-SNOM images taken at taken at 6.9 μm excitation wavelength show nano-confined near-field polaritonic hotspots (Fig. 2g). We tune then the field distribution by exposing the sample to hydrogen as described above and show in Fig. 2i and 2k and decrease the hotspot intensity by a factor of 2 (Fig. 2n green color). We then further tune the intensity by erasing the writing using c-AFM negative voltage on top of the hBN and decrease the field intensity further (Fig. 2i and purple line in Fig 2n). Nanoscale reconfigurability of the local conductivity in correlated oxides enables manipulation of sub-diffraction light-matter interactions and unique opportunities to control propagating nano-confined polariton fields. These results demonstrate a novel route to design a new class of cavities where multimodal interference of HPhPs can enable tunable and reconfigurable polaritonic hotspots characterized by ultra-high field confinement and enhancement in nano-confined modal volumes.

This is interesting potentially impactful work. I believe this paper could be publishable in NCOMMS, but the manuscript needs extensive revision and a little change in focus.

Author Response #4:

We thank the Reviewer for valuable and encouraging comments and remarks.

Reviewer #3 (Remarks to the Author):

The paper present a novel way to control polaritons in van der Waals materials using a SNO substrate which can be written by locally applying voltage with an AFM tip.

The method is very interesting due to the higher resolution compared with phase change materials, and timely. The paper is technically correct to the best of my knowledge. However there are some minor points to be addressed before acceptance

Author Response:

We thank the Reviewer for their valuable and encouraging comments and remarks.

1) There are some minor typos such as "A tip with a large positive bias remove oxygen"

Author Response #1:

This sentence is corrected to the following new one on page 5 of the manuscript.

A biased tip enables oxygen vacancy generation on the SNO surface, providing a controllable route to manipulate the surface conductivity of SNO.

2) About the sentence in the introduction:

"So far, such tunable polaritons have relied on temperature⁸⁻¹¹ modulation, which introduces loss, as the phonon lifetime is reduced when the temperature, and correspondingly the phonon-phonon scattering events, increase. In addition, this route suffers from inaccessibility of nanoscale manipulation due to the inability to design localized thermal manipulation. As a result, reconfigurability of nanopolaritonic structures remains limited."

This sentence is misleading, since several references achieved polariton tuning without the need to maintain high temperature, and hence without impact on the polariton quality. This includes ref. 7 based on GST, and the method of intercalation of MoO₃ in ref 33. The Authors should revise this sentence to clarify this point.

Author Response #2:

Following referee advice, we have modified this sentence to:

Tunable polaritons generation that relies on temperature¹⁻⁴ modulation, introduces loss, as the phonon lifetime is reduced when the temperature, and correspondingly the phonon-phonon scattering events, increase and potentially may also limit the spatial scale. Advancing methods to reconfigure nanopolaritonic structures is therefore desirable.

3)

Related to the previous point: an additional difference that should be pointed out is that phase change materials can be written and erased after the 2D materials are transferred shining light through them, while here the c-AFM tip method needs to be done prior to the transfer of the van der Waals flake and is, as far as I understand, irreversible. This point should also be mentioned in the introduction to better relate this work with existing literature.

Author Response #3:

We thank the Reviewer for this comment. We agree with the Reviewer, we have shown an advantage of using SNO platform for nanophotonics because of the ability to reversibly tune the pattern surface thereby tuning polaritons via writing with a positive tip-bias and erasing a region using a negative bias. This write/erase mode can be applied either on bare SNO surface or on hBN/SNO surface. To that end we wrote a square pattern on pristine SNO by applying 8V potential at the c-AFM tip (see Fig. S4 a, Topography b, SKPM surface potential image). We then exfoliated a 45 nm thick hBN, followed by erasing several circular patterns by applying -10V potential at the c-AFM tip (cyan dash boxes, Fig. S4 e) and -8V potential at the c-AFM tip (cyan dash circle, Fig. S4 f) on top of hBN. This demonstrates the write/erase ability on the heterostructure hBN/SNO platform. A similar different pattern write/erase patterns on hBN/SNO heterostructure are also shown in Fig. S5. We have added these figures on the SI and corresponding changes on the manuscript since this question was asked by Reviewer 1 above.

4) About the sentence:

"Unlike most metals, for correlated oxides like SNO, increasing temperature or disorder does not hasten electron scattering processes; instead, increasing the temperature increase conductivity 40"

It would be helpful to understand if this phenomenon is due to increased mobility or increased carrier's density.

Author Response #4:

We thank the reviewer for this comment. Hall effect measurements on SmNiO_3 ², have shown that the increase in conductivity with temperature in the insulating phase is mainly due to the increased mobility.

5) Unless I am missing it, I cannot find the values (measured or estimated) of the permittivity / refractive index of the SNO in the two states (referred to in the supplementary as epsilon_1, and epsilon_2). These values should be provided.

Author Response #5:

We thank the reviewer for noticing this overlook in our part. The refractive indices were taken from literature³

On page 9 of the SI we have added this reference and modify the following sentence with a reference:

"... ϵ_1 and ϵ_2 are the permittivity of SNO in insulator and pristine phase⁶..."

6) Please clarify in the caption of figure 4 which panels are experiments and which are simulations

Author Response #6:

Done.

References

- 1 Britnell, L. *et al.* Electron tunneling through ultrathin boron nitride crystalline barriers. *Nano Lett* **12**, 1707-1710, doi:10.1021/nl3002205 (2012).
- 2 Ha, S. D. *et al.* Hall effect measurements on epitaxial SmNiO₃ thin films and implications for antiferromagnetism. *Physical Review B* **87**, 125150, doi:10.1103/PhysRevB.87.125150 (2013).
- 3 Li, Z. *et al.* Correlated Perovskites as a New Platform for Super-Broadband-Tunable Photonics. *Advanced Materials* **28**, 9117-9125, doi:<https://doi.org/10.1002/adma.201601204> (2016).

REVIEWER COMMENTS

Reviewer #1 (Remarks to the Author):

I have carefully reviewed the revised manuscript. I think the authors have done a good job in addressing the comments and questions raised by me and other referees. In view of their efforts and the proper revisions, I suggest acceptance of their manuscript. However, I still insist on modifying the word "metasurface" throughout the manuscript. A metasurface should be able to tailor the wavefront of the incident light and control its spatial flow. The architecture reported here has no similar functions. Nevertheless, this modification is not mandatory.

Reviewer #2 (Remarks to the Author):

Review: Aghamiri et al.

Again, this work attempts to show that one can potentially manipulate polariton dynamics by controlling the electronic properties of an underlying SNO correlated Oxide substrate. The SNO can be readily patterned allowing tailoring of the PhPs of the overlain material. This is an important aspect of Polaritonic Device engineering. The work is noteworthy, though other Modification paradigms have been explored using a material like SNO is new territory. My concern continues to be the manuscript needs to do a better job of illustrating Polariton control. Authors figure 3c does this in part, but the illustration could be more definitive. The modifications to figure 2 don't quite illustrate the control as well as could be. The authors could simply plot the figure in 3c as a function of voltage of the underlying SNO. This would show the control well.

Again, showing that one can generate s-SNOM contrast from patterned substrates (as shown in figure 2) is expected, but a far cry from "Polariton Control". Polaritonic mapping of the modified underlying substrate is well understood. It is straight forward to show direct impact on polariton propagation. An actual measure of the change in Polariton dispersion, (wavevector or velocity), properties as a function of voltage would make this manuscript stronger. Figure 2p may do this, but it's not clear or well described. For example, it not clear if the figure is for hBN/SNO or α -MoO₃/SNO. (Text seem different from figure caption.) For an additional example, the Authors could/should simply pattern a metasurface to show Polariton focusing or show directly the spatial propagation changes in dispersion induced by the patterning used. Again, characterization of the dispersion changes induced by the substrate are key. Illustrating functionality would make this work much stronger. In general, the claims are well supported. The better illustration of functionality would make this manuscript adequate to publish.

I recommend that the manuscript be accepted after modest revision.

Reviewer #3 (Remarks to the Author):

The Authors' response and the new version of the manuscript addressed my concerns and I believe that the paper is now ready for publication.

-----Authors Responses to Reviewers Comments are *italicized*-----
and
-----Authors addition to the manuscript and SI are **highlighted in yellow**-----

Reviewer #1 (Remarks to the Author):

I have carefully reviewed the revised manuscript. I think the authors have done a good job in addressing the comments and questions raised by me and other referees. In view of their efforts and the proper revisions, I suggest acceptance of their manuscript. However, I still insist on modifying the word “metasurface” throughout the manuscript. A metasurface should be able to tailor the wavefront of the incident light and control its spatial flow. The architecture reported here has no similar functions. Nevertheless, this modification is not mandatory.

Reviewer #3 (Remarks to the Author):

The Authors' response and the new version of the manuscript addressed my concerns and I believe that the paper is now ready for publication.

Author Response:

We thank Reviewer #1 and Reviewer #3 for all the important comments and questions that certainly improved the manuscript.

Reviewer #2 (Remarks to the Author):

Review: Aghamiri et al.

Again, this work attempts to show that one can potentially manipulate polariton dynamics by controlling the electronic properties of an underlying SNO correlated Oxide substrate. The SNO can be readily patterned allowing tailoring of the PhPs of the overlain material. This is an important aspect of Polaritonic Device engineering. The work is noteworthy, though other Modification paradigms have been explored using a material like SNO is new territory.

Author Response:

We thank the Referee for recognizing the value of our work and very valuable questions/suggestions and positive comments.

My concern continues to be the manuscript needs to do a better job of illustrating Polariton control. Authors figure 3c does this in part, but the illustration could be more definitive. The modifications to figure 2 don't quite illustrate the control as well as could be. The authors could simply plot the figure in 3c as a function of voltage of the underlying SNO. This would show the control well.

Again, showing that one can generate s-SNOM contrast from patterned substrates (as shown in figure 2) is expected, but a far cry from “Polariton Control”. Polaritonic mapping of the modified underlying substrate is well understood. It is straight forward to show direct impact on polariton propagation. An actual measure of the change in Polariton dispersion, (wavevector or velocity), properties as a function of voltage would make this manuscript stronger. Figure 2p may do this, but it's not clear or well described. For example, it not clear if the figure is for hBN/SNO or α -MoO₃/SNO. (Text seem different from figure caption.)

Author Response:

We thank the Referee for this observation we overlooked. We have modified the figure caption in Fig. 2p as shown below and indicated in the main manuscript on page 11 that the calculation is done for α -MoO₃:

Fig. 2p

p, Ratio of polariton wavevector to free space photon wavevector vs voltage and temperature for α -MoO₃.

For an additional example, the Authors could/should simply pattern a metasurface to show Polariton focusing or show directly the spatial propagation changes in dispersion induced by the patterning used. Again, characterization of the dispersion changes induced by the substrate are key. Illustrating functionality would make this work much stronger. In general, the claims are well supported. The better illustration of functionality would make this manuscript adequate to publish.

I recommend that the manuscript be accepted after modest revision.

Author Response:

We thank the Referee for this insightful suggestion. As the Reviewer indicated, the combination of Fig. 1h&i, Fig. 2o&p or Fig. 3c already elucidate HPhP characteristics as a function of the dielectric function of SNO. The index change can be caused by voltage, H₂ doping or temperature changes which would modify the HPhP dispersion. We demonstrate the role of the applied writing voltage, which modifies the substrate complex refractive index (Fig. 1h&i, Fig. 2o&p), on the hyperbolic polariton wavevector, by showing in the calculation plot below (extension of Fig. 3c). The figure shows the dispersions of HPhPs in hBN with 60nm thickness, with the SNO (no H₂) at writing voltages 7 V and 10 V. Here, we assume that the dielectric constant of SNO changes linearly with the applied voltages (Fig. 1h&i, Fig. 2o&p), which could be a good approximation considering the contrast change with voltage (Fig. 1h&i, Fig. 2o&p).

We have also added a new SI figure, Fig. S4 to show the patterning voltage effects and spatial changes of the nano-confined near-field polaritonic hotspot intensity (*s*-SNOM amplitude contrast) dispersion induced by the patterning used (two different voltages, 8V and 10V). The confined modes change in intensity profile with changing excitation wavelength and visible voltage dependence changes (particularly close to resonance wavelength $\sim 7\mu\text{m}$) which further tunes the mode spatial intensity distribution. We have also re-numbered SI figure numbers accordingly.

We have indicated the addition of this figure in the main text on page 11 as:

(see Fig. S4 for more wavelength and writing voltage dependence images).

Fig. S4 | S-SNOM second harmonic near-field amplitude, s_2 , images of pristine SNO with conductive writing patterns made by applying potential at the c-AFM tip, 10 V (three top rows) and 8 V (three bottom rows in a-h) with a hBN flake transferred on top of the pattern taken at laser excitation wavelength **a**, $\lambda = 6.35 \mu\text{m}$, **b**, $\lambda = 6.4 \mu\text{m}$, **c**, $\lambda = 6.45 \mu\text{m}$, **d**, $\lambda = 6.9 \mu\text{m}$, **e**, $\lambda = 6.95 \mu\text{m}$, **f**, $\lambda = 7 \mu\text{m}$, **g**, $\lambda = 7.05 \mu\text{m}$, **h**, $\lambda = 7.1 \mu\text{m}$. **i**, Normalized amplitude intensity plots of nano-confined hot-spots $s_2(\text{writing})/s_2(\text{hBN-SNO})$ as a function of wavelength. Scale bars indicate $1 \mu\text{m}$.

REVIEWERS' COMMENTS

Reviewer #2 (Remarks to the Author):

I believe the authors have adequately addressed my concerns with their additional changes.

I recommend the manuscript for publication.